# Mental Well-Being of Czech University Students: Academic Motivation, Self-Compassion, and Self-Criticism

**DOI:** 10.3390/healthcare10112135

**Published:** 2022-10-27

**Authors:** Yasuhiro Kotera, Sarah Maybury, Gillian Liu, Rory Colman, Jenai Lieu, Jaroslava Dosedlová

**Affiliations:** 1School of Health Sciences, University of Nottingham, Nottingham NG7 2HA, UK; 2College of Health, Psychology and Social Care, University of Derby, Derby DE22 1GB, UK; 3Department of Psychology, Masaryk University, 602 00 Brno, Czech Republic

**Keywords:** Czech university students, self-compassion, intrinsic motivation, self-criticism, amotivation, self-hate, self-inadequacy

## Abstract

University students in the Czech Republic suffer from a low level of mental well-being. Research in other university student populations suggests that academic motivation, self-compassion, and self-criticism are strongly related to mental well-being. Students who are motivated to study, are kind toward themselves, and are less judgmental of themselves tend to have a high level of mental well-being. These relationships had not been evaluated in Czech students. Accordingly, this cross-sectional study aimed to evaluate the relationships between mental well-being, academic motivation (intrinsic motivation, extrinsic motivation, and amotivation), self-compassion (self-reassurance) and self-criticism (self-inadequacy and self-hate). Of 130 students approached, a convenience sampling of 119 psychology students at a university in the Czech Republic completed a survey regarding these constructs. Correlation, regression, and path analyses were conducted. Mental well-being was positively associated with intrinsic motivation and self-compassion, and negatively associated with amotivation and self-criticism. Self-compassion was identified as the strongest predictor of mental well-being. Lastly, intrinsic motivation mediated the pathway from self-compassion to mental well-being, but not the one from self-inadequacy to mental well-being, and the one from self-hate to mental well-being. Our findings can help educators to identify effective means to protect students’ mental well-being. Cultivating students’ self-compassion may be helpful to protect their mental well-being. University staff and educators in the Czech Republic need to consider ways to embed self-compassion training into their students’ programmes or university life.

## 1. Introduction

### 1.1. Poor Mental Well-Being of Czech Students

The promotion of mental well-being is at the forefront of many government agendas, due to the social, economic, and human burden that accompanies poor mental well-being [1]. Although mental well-being is a global health concern, the approach toward and experience of mental well-being, as well as the reception of mental health interventions, differ between countries, due to unique historical and political considerations [2]. For example, in ex-Soviet countries, the treatment of individuals with mental health problems was highly stigmatised, discriminatory, and often abusive [3]. With the fall of the Soviet Union, came a shift in the classification of mental health, from models established by Soviet health institutions to Western psychiatric diagnostic categories [4]. Although this shift brought a change in practices, the negative attitudes toward mental health persist, jeopardising effective interventions, and leading to worsened mental health outcomes [5]. This lasting negative perception towards mental health has resulted in a treatment gap and insufficient resources to aid those living with mental disorders [6]. Moreover, these attitudes have been internalised by those suffering from mental illness, who demonstrate feelings of shame, resulting in additional barriers to treatment-seeking behaviours [7].

In the Czech Republic, mental health expenditure amounts to approximately 4% of total health expenditure, resulting in insufficient provision of services and psychiatrists with extremely unmanageable patient loads [8]. In a country where an estimated one in five person suffers from an affective, anxiety, or alcohol- or substance-abuse disorder, the absence of adequate care for these individuals often results in further deterioration in already neglected members of society [6]. Rates of alcohol dependency in the Czech Republic are almost double those in Central Europe, with the highest prevalence found in individuals aged 18 to 29, the common age of undergraduate university students [9]. University students are at risk for further mental health problems, which reach their peak in young adulthood, often exacerbated by the academic and social pressures associated with the transition to university [10]. 

Additionally, the COVID-19 pandemic has presented additional stressors to Czech university students. The way the Czech government disseminated information in the initial stages of the pandemic led to mass psychological traumatisation of the general population [11]. Combined, these factors highlight the need for attention and research into ways to improve mental well-being for university students in the Czech Republic [12].

### 1.2. Academic Motivation and Mental Well-Being

One well-established avenue for this improvement has been through enhancing academic motivation, as understood through the lens of the self-determination theory (SDT) [13]. SDT proposes that all humans possess three primary needs: relatedness (feelings of belonging and connection to others), autonomy (self-directed action to exercise one’s will), and competence (the effective interaction with one’s surroundings to build capabilities). If these needs are unmet, a person may feel their life lacks meaning, which can result in depressive symptoms and suicidal feelings [14]. 

A fundamental aspect of SDT is motivation, which exists on a continuum and includes intrinsic motivation, extrinsic motivation and amotivation. Intrinsic motivation encourages individuals to seek opportunities to learn, explore and be challenged based on a task itself being meaningful, and results in positive outcomes in university students when it comes to stress, psychological health, and academic performance [15]. Extrinsic motivation is driven by the need to meet an external demand or obtain rewards, and amotivation refers to the absence of an intention to act [13], which are both associated with poorer academic and mental well-being outcomes [16]. 

Although SDT claims universal applicability, most studies about motivation have been performed in the United States and other Western, educated, industrialised, rich, and democratic (WEIRD) nations [17]. Some argue that it employs an individualist conception of people due to the focus on autonomy that is not relevant across cultures [18]. However it is argued that SDT acknowledges autonomous interdependence, which aligns with more collectivist-based value systems [19]. SDT has been shown to be an effective model to investigate motivation across different societies, given that cultural nuances are considered and applied to the population being studied [20,21]. 

Although SDT has demonstrated its suitability to study non-WEIRD cultures, there remains a need to grow the body of research into the specificities of ex-Soviet states, including the Czech Republic, as cultural differences are important to understanding psychological outcomes [22]. Rather than a static set of shared practices that precedes psychological phenomena, culture cultivates a person’s worldview, which is integrated into their psyche and is fundamental to the understanding of processes such as motivation [23].

### 1.3. Self-Compassion and Self-Criticism

Studies across cultures provide a wealth of evidence that self-compassion and self-criticism are also related to mental well-being. Comparative studies between a range of countries have shown self-compassion to be a valid construct [24] benefiting diverse populations either measured as a trait [25] or following intervention to enhance self-compassion [26]. Evidence of negative mental well-being implications from self-criticism are comparably robust [27], with students at risk from its psychopathological effects [28]. To date, there is little information on the role of self-compassion and self-criticism in students within the Czech Republic, indicating a requirement for research into how these psychological factors contribute to their mental well-being.

Gilbert’s biopsychosocial theory conceptualises self-compassion and self-criticism as originating in distinct organisations of motivation, emotion, and behaviour corresponding to evolutionary social relating modes, referred to as social mentalities [29]. Self-compassion is a caring social mentality, corresponding with neurophysiological systems promoting rest and digest activities [30]. Self-criticism on the other hand is a competitive social mentality, corresponding with threat-based monitoring and fight–flight neurophysiological systems [29]. At times of stress or setback, individuals experience emotion and exhibit behaviour related to how these systems are activated. Those able to behave self-compassionately and with self-reassurance, are afforded protection from negative psychological consequences, whereas self-critical individuals lack such protection and may even exacerbate the negative consequences by self-attacking [31]. Self-criticism can take the forms of self-inadequacy, a sense of personal inadequacy and disappointment with self, and self-hate, a sense of self-disgust and self-hatred with intentions of self-harm [31]. Within non-Czech student populations, each of these forms of self-criticism are shown to be associated with poorer mental well-being [31,32,33,34,35].

### 1.4. Intrinsic Motivation as Mediator for Self-Compassion to Mental Well-Being

Gilbert’s model regards motivation as determining the social mentality one adopts to cope with challenges. For instance, functioning to protect from possible harm to social standing, the self-critical process is motivated by underlying fears of social disconnection and shame [29]. Self-criticism has been shown to be associated with extrinsic motivation [36,37,38], hindering academic goal progress [39]. Driven by perceived threat, self-criticism stimulates defensive and defeatist behaviours with corresponding negative effects [40] which can manifest in amotivation. A self-reassuring process on the other hand is motivated by caring for self, rather than social comparison, fostering a sense of self-efficacy and courage [30,41]. 

Based on attachment research findings, self-reassurance is heorized to develop through experiences of a secure and reassuring environment. Indeed, students with secure attachment styles have been found most able to self-reassure [42]. Leak and Cooney found intrinsic motivation to mediate between a secure attachment style and psychological well-being when investigating adult student romantic relationships [43]. What is not yet established, however, is how self-reassurance and intrinsic motivation interact to influence mental well-being. As findings in university students of different cultures indicate self-compassion to have positive association with intrinsic motivation [44,45,46,47,48], it is hypothesised in this study that intrinsic motivation may mediate the relationship between self-compassion and mental well-being in Czech students. Because interventions to increase self-compassion in students have provided evidence of enduring mental well-being effects up to six months following intervention [49], self-compassion has promise as an ability that can be developed for Czech students. By ascertaining the relationships between self-compassion, self-criticism, motivation, and mental well-being, recommendations can be made on the most appropriate means to improve Czech students’ mental well-being.

### 1.5. Study Aims

This study aimed to understand the mental well-being of Czech university students. Therefore, the objectives were (1) to evaluate the relationship among mental well-being, academic motivation (intrinsic motivation, extrinsic motivation, and amotivation), self-compassion, and self-criticism; (2) to identify predictors of mental well-being; and (3) assess whether intrinsic motivation mediates the pathway from self-compassion/self-criticism to mental well-being.

## 2. Materials and Methods

### 2.1. Participants

Opportunity sampling was used to recruit 130 undergraduate students in psychology at a Czech university in Brno, of which 119 completed the measures. The students were Czech (*n* = 98) and Slovakian (*n* = 21), had an age range between 19 and 44 years (M = 21.87, SD = 3.32), and consisted of 93 females and 20 males. This sample reflects the demographic of the general population of psychology students in the Czech Republic [50]. Students were recruited through announcements given by programme tutors. Overall, the number of participants reached the required sample size of 115 according to statistical power calculations (84: two tails, *p*H1 (*r*) = 0.30 medium [51], α = 0.05, Power = 0.80, *p*H0 = 0 [52]). Participants provided consent to their engagement in the study.

### 2.2. Materials

The study evaluated the relationship between mental well-being, academic motivation (intrinsic motivation, extrinsic motivation and amotivation), self-compassion (self-reassurance), and self-criticism (self-inadequacy and self-hate) through a survey regarding these constructs.

The Self-Criticising/Attacking and Self-Reassuring Scale (FSCSR) was used to assess self-criticism and self-reassurance in participants, by looking at how they think of themselves in adverse circumstances [31]. The FSCSR consists of 22 items scored on a five-point Likert scale (0 = “Not at all like me” to 4 = “Extremely like me”). The scale that contains three dimensions: two forms of self-criticism (self-inadequacy and self-hate) and one form of self-reassurance. Self-inadequacy refers to a belief of personal deficiency (e.g., “I am easily disappointed in myself”), self-hate refers to a desire to punish the self (e.g., “I have a sense of disgust toward myself”), and self-reassurance refers to the ability to soothe oneself when things go wrong (e.g., “I am able to remind myself of positive things about myself”). The FSCSR subscales are known for their reliability (α = 0.90 for self-inadequacy, α = 0.86 for self-hate, and α = 0.86 for self-reassurance) and validity (|*r*| = 0.45–0.77) [31].

The Academic Motivation Scale (AMS) [53] measures three variations of motivation that are categorised into seven subtypes: (i) amotivation, (ii) extrinsic motivation, and (iii) intrinsic motivation. Amotivation refers to the lack of self-determination (e.g., “I can’t understand what I am doing in school”), extrinsic motivation refers to external and identified factors (e.g., “Because I want to have ‘the good life’ later on), and intrinsic motivation is the sense of knowing or accomplishment from the action (e.g., “Because I experience pleasure and satisfaction while learning new things”). The scale consists of 28 items on a seven-point Likert scale (1 = “Does not correspond at all” to 7 = “Corresponds exactly”). The AMS is demonstrated to have adequate to high internal consistency (α = 0.62–0.91) [53].

The Short Warwick–Edinburgh Mental Well-Being Scale (SWEMWBS) [54] evaluated mental well-being by using the shortened version of the original 14-item scale to the seven-item scale [55]. The SWEMWBS is measured on a five-point Likert scale (1 = “None of the time” to 5 = “All of the time”). The scale has been demonstrated to be mostly free of item bias [54] and is considered well-established to appraise mental well-being [56]. Participants completed the survey based on their experiences within the past two weeks regarding hedonic and eudaimonic well-being (e.g., “I’ve been dealing with problems well”). The SWEMWBS is regarded to have high internal consistency (α = 0.85) [54].

### 2.3. Analysis

A cross-sectional design was used to conduct correlation, regression, and path analyses in the study. The data was screened for outliers and the assumptions of parametric tests. Afterward, correlations between mental well-being, academic motivation, self-compassion, and self-criticism were evaluated by using IBM SPSS version 27. Lastly, path analyses were conducted to analyse how self-criticism and self-reassurance may mediate in the relationship between mental well-being and academic motivation by using the Process Macro 3 for SPSS [57] with 5000 bootstrapping re-samples and bias-corrected 95% confidence intervals (CIs) for indirect effects were applied.

### 2.4. Ethics

The research ethics were approved by the researcher, JD’s university research ethics committee. Participants provided their consent and were informed of their rights including their right to withdraw from the study.

## 3. Results

Descriptive statistics of all variables are presented in Table 1. High internal consistency was demonstrated for all variables (α = 0.73–0.90).

### 3.1. Correlation (Objective 1)

Pearson’s correlation was performed to evaluate the relationship between mental well-being, academic motivation (intrinsic motivation, extrinsic motivation, and amotivation), self-reassurance, and self-criticism (self-inadequacy and self-hate) (Table 2). Point biserial correlations were calculated for gender (0 = female, 1 = male).

Mental well-being was positively associated with intrinsic motivation and self-reassurance, and negatively associated with amotivation, self-inadequacy and self-hate. None of the three motivation types were interrelated. Self-reassurance was positively associated with intrinsic motivation and negatively associated with amotivation, self-inadequacy, and self-hate. Self-inadequacy and self-hate were positively related to each other. 

### 3.2. Regression (Objective 2)

Multiple regression analyses were conducted to identify predictors of mental well-being (outcome variable). Significant correlates with mental well-being, namely intrinsic motivation, amotivation, self-reassurance, self-inadequacy, and self-hate were entered as a predictor variable. The adjusted coefficient of determination (Adj. R^2^) was calculated to determine the degree of variance in the population (Table 3). Multicollinearity was not a concern (variance inflation factors < 10).

These five predictor variables were all significant, and predicted 48% of the variance in mental well-being altogether, indicating a large effect size [51]. Self-reassurance had the greatest impact on mental well-being (β = 0.46). 

### 3.3. Mediation (Objective 3)

Three models of path analyses were conducted to assess whether intrinsic motivation would mediate the pathways from self-compassion/self-criticism to mental well-being. Model 4 in the process macro (parallel mediation model) [57] was used. Mental well-being was entered as an outcome variable, intrinsic motivation as a mediator variable, and self-reassurance, self-inadequacy, and self-hate as a predictor variable.

First, intrinsic motivation partially mediated the pathway from self-reassurance to mental well-being (Figure 1a). All pathways including the direct effect from self-reassurance to mental well-being, and total effect that includes the impact of intrinsic motivation, were significant (* *p* < 0.05; ** *p* < 0.01; *** *p* < 0.001). When self-reassurance increases mental well-being, it also impacts intrinsic motivation, which in turn improves mental well-being.

Second, intrinsic motivation did not mediate the pathway from self-inadequacy to mental well-being (Figure 1b). Although the direct effect from self-inadequacy to well-being, the total effect that includes the impact of intrinsic motivation on mental well-being, and intrinsic motivation’s effect on mental well-being were significant, a pathway from self-inadequacy to intrinsic motivation was not significant. 

Lastly, intrinsic motivation did not mediate the pathway from self-hate to mental well-being either (Figure 1c). Although the direct effect of self-hate on well-being, the total effect that includes the impact of intrinsic motivation on mental well-being, and intrinsic motivation’s effect on mental well-being were significant, a pathway from self-hate to intrinsic motivation was not significant.

## 4. Discussion

This study evaluated the relationship between mental well-being, academic motivation, self-compassion (self-reassurance) and self-criticism (self-inadequacy and self-hate) in university students in the Czech Republic. Mental well-being was positively associated with intrinsic motivation and self-compassion, and negatively associated with amotivation and self-criticism. Self-compassion was identified as the strongest predictor of all of mental well-being. Lastly, intrinsic motivation mediated the pathway from self-compassion to mental well-being but did not mediate the pathways from self-criticism to mental well-being. 

Consistent with previous research conducted in other cultures, intrinsic motivation and self-compassion were associated with mental well-being [15,44]. The significant correlation of mental well-being with intrinsic motivation and self-compassion in this study showed these may be universal regardless of culture differences. The positive association of mental well-being and intrinsic motivation in Czech students supports SDT as an explanatory theoretical model. Extending previous findings in Asia and Africa [46,58], the present study showed the applicability of SDT to non-WEIRD cultures that are located in Europe. 

Self-compassion was also a strong predictor of mental well-being in the Czech students. This is consistent with previous findings from different populations [49,59,60]. Self-compassion interventions have been put into practice, and achieved desirable outcomes, such as increases in self-compassion, optimism, self-efficacy, and happiness and decreases in depression, anxiety, and stress [61,62,63,64]. Because of these outcomes, implementing self-compassion training appears to be a useful approach for better self-care and mental well-being [65,66]. Supporting self-compassion may be a less stigmatised approach for better mental well-being, instead of focusing on reduction of mental distress [67,68]. Because there is a negative perception toward people with mental health problems in Czech higher education [5], this significant relationship between self-compassion and mental well-being may help identify an effective means to support the mental well-being of Czech students. 

Lastly, intrinsic motivation mediated the pathway from self-compassion to mental well-being, but did not mediate the pathways from self-criticism to mental well-being. When self-compassion impacts positively on mental well-being, there is an indirect positive impact of intrinsic motivation on mental well-being too. Associations between self-compassion and intrinsic motivation may play an important role in this pathway [69,70]. To be intrinsically motivated and highly autonomous, one needs to feel safe and content (i.e., self-compassionate) [71]. This mediation model suggests that by cultivating self-compassion, not only the Czech students’ mental well-being, but also their intrinsic motivation can be enhanced. In other words, educators in the Czech universities can teach students to be kind and understanding toward themselves for mental well-being purposes, which can also help students to be more autonomous. The self-compassion training can be embedded in the curriculum. Alternatively, for students who feel it challenging to practice self-compassion exercises such as meditation and imageries [72,73], forest bathing may be recommended to support self-compassion [74,75]. 

Intrinsic motivation, however, did not mediate the self-criticism to mental well-being pathways. When self-criticism damages mental well-being, it does not impact intrinsic motivation positively nor negatively. This highlights the non-significant association between self-criticism and intrinsic motivation [76]. Furthermore, Powers et al. found no association between self-criticism and intrinsic motivation, suggesting that self-criticism and autonomy may independently contribute to mental well-being [39]. This may highlight the fact that self-criticism needs to be mitigated independent from motivation in order to protect mental well-being [77]. As noted earlier, self-compassion training may be helpful as it can reduce self-criticism [78]. 

Several limitations should be noted. First, the generalisability of our findings might have been compromised, as our recruitment was through opportunity sampling at one university. Furthermore, although the sample satisfied the required size, a larger sample could have enhanced the generalisability of the findings. Secondly, the two scales used in this study were rather long (FSCSR 22 items and AMS 28 items). Shorter versions [79,80] could have been used to reduce the workload of the students. Lastly, the causal directions of these relationships were not evaluated. Longitudinal data would help inform the causality, which may help develop more effective approaches. 

## 5. Conclusions

This study evaluated the relationships among mental well-being, academic motivation, self-compassion, and self-criticism in university students in the Czech Republic. Consistent with findings from other student populations, strong relationships among mental well-being, self-compassion, and intrinsic motivation were identified in the Czech students. Cultivating self-compassion may enhance mental well-being as well as intrinsic motivation. Our findings can help educators to develop effective approaches to support the mental well-being among Czech university students. 

## Figures and Tables

**Figure 1 healthcare-10-02135-f001:**
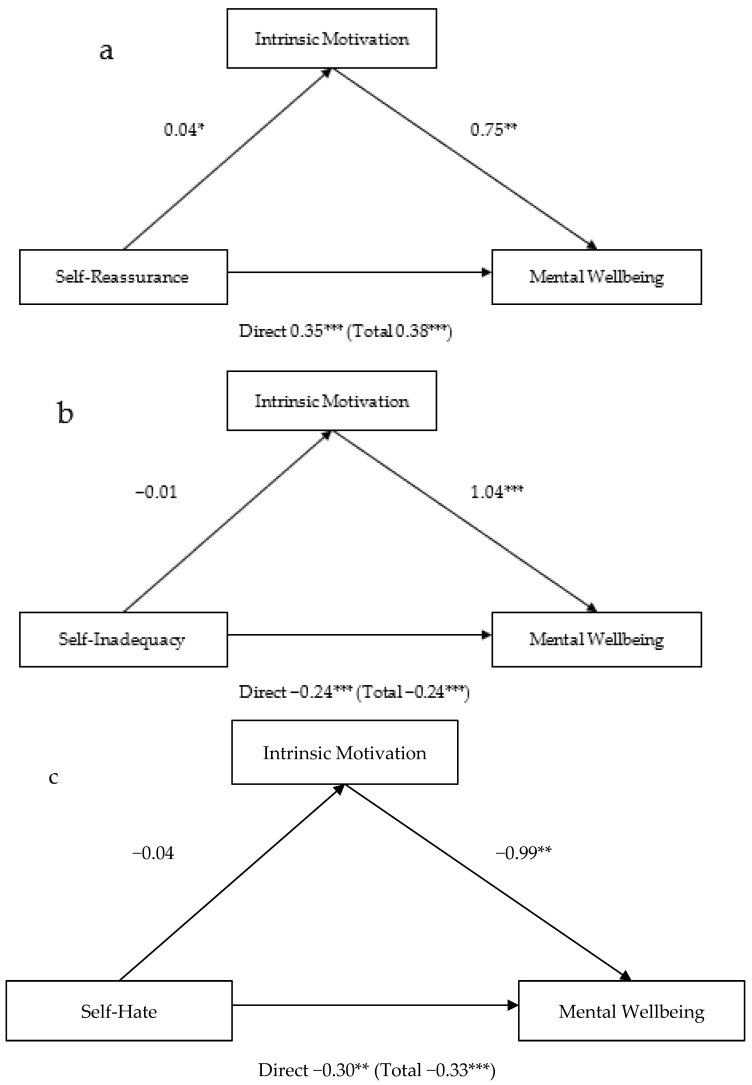
Three models of path analyses with intrinsic motivation as mediator. * *p* < 0.05; ** *p* < 0.01; *** *p* < 0.001, Direct = direct effect; Total = total effect. Mediation model of intrinsic motivation on the (**a**) self-reassurance to mental wellbeing pathway, (**b**) self-inadequacy to mental health pathway, and (**c**) self-hate to mental wellbeing pathway.

**Table 1 healthcare-10-02135-t001:** Descriptive statistics.

	M	SD	α
Gender	93 females, 20 males, 6 unanswered		
Age	21.87	3.32	
Mental well-being	25.21	3.80	0.73
Intrinsic motivation	4.55	1.10	0.90
Extrinsic motivation	4.97	1.02	0.83
Amotivation	1.52	0.88	0.80
Self-reassurance	20.35	6.08	0.85
Self-inadequacy	17.35	8.12	0.88
Self-hate	3.92	4.20	0.82

**Table 2 healthcare-10-02135-t002:** Correlations among mental well-being, academic motivation, self-compassion, and self-criticism.

		1	2	3	4	5	6	7	8	9
1	Gender (0 = F, 1 = M)	-								
2	Age	0.25 **	-							
3	Mental Well-being	0.05	0.13	-						
4	Intrinsic motivation	0.12	−0.09	0.33 **	-					
5	Extrinsic motivation	−0.27 **	−0.11	−0.01	0.18	-				
6	Amotivation	0.05	−0.03	−0.41 **	−0.17	0.03	-			
7	Self-reassurance	0.03	−0.06	0.61 **	0.21 *	−0.03	−0.33 **	-		
8	Self-inadequacy	−0.10	−0.10	−0.52 **	−0.06	0.22 *	0.49 **	−0.63 **	-	
9	Self-hate	−0.02	−0.01	−0.37 **	−0.14	0.05	0.51 **	−0.60 **	0.73 **	-

* *p* < 0.05, ** *p* < 0.01.

**Table 3 healthcare-10-02135-t003:** Multiple regression analyses for mental well-being.

	B	SE_B_	β	95% CI (Low, Up)	
Intrinsic motivation	0.76	0.24	0.22 **	0.27	1.23
Amotivation	−0.91	0.34	−0.21 **	−1.60	−0.23
Self-reassurance	0.29	0.06	0.46 ***	0.18	0.40
Self-inadequate	−0.15	0.05	−0.32 **	−0.25	−0.05
Self-hate	−0.26	0.09	−0.28 **	−0.07	−0.44
Adj. R^2^	48%				

Outcome variable: Mental well-being. B = unstandardised coefficients; SE_B_ = standard error of the coefficient; β = standardised coefficients. ** *p* < 0.01; *** *p* < 0.001.

## Data Availability

The data presented in this study are available on request from the corresponding author. The data are not publicly available due to ethical restrictions.

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
