# Peer review of "Mental Well-Being of Czech University Students: Academic Motivation, Self-Compassion, and Self-Criticism"

_healthcare, 2022, doi:10.3390/healthcare10112135_

Round 1

Reviewer 1 Report

Remarks:

1. The theoretical background of the research is narrowly outlined.

2. The research sample was too small, which is a serious drawback of the research.

3. The method, technique, tool should be presented more precisely.

4. The selection of the test sample should be accurately presented.

5. The discussion of the results is poorly written.

1. “Accordingly, this cross-sectional study aimed to evaluate the relationships between mental wellbeing, academic motivation (intrinsic motivation, extrinsic motivation and amotivation), self-compassion (self-reassurance) and self-criticism (self-inadequacy and self-hate)”.

2. “Our findings can help educators to identify effective means to protect students’ mental wellbeing. Cultivating students’ self-compassion may be helpful to protect their mental wellbeing”. I can agree with that.

3. Researchers conducted research in the Czech Republic. They represent different centers, therefore they can test a larger number of people. There are more places in the Czech Republic where you can do research.

4. The authors selected too small a sample of respondents for the research, therefore the results and conclusions cannot be generalized (130 – 119). Statistical analysis on such a small sample is not correct.

5-6. Conclusions cannot be generalized. Too small sample was tested.

Author Response

Response Letter

Manuscript ID: healthcare-1948142

"Mental wellbeing of Czech university students: academic motivation, self-compassion and self-criticism”

Dear Reviewers,

Thank you for your helpful feedback. We have systematically revised our manuscript addressing the points you have raised. Please see our responses below. We hope this revised paper is now acceptable for publication. We extend our sincere gratitude to you for your feedback that has significantly helped to strengthen the paper.

Reviewer 1

Reviewer 1’s comment 1

  1. The theoretical background of the research is narrowly outlined.

  1. The research sample was too small, which is a serious drawback of the research.

  1. The method, technique, tool should be presented more precisely.

  1. The selection of the test sample should be accurately presented.

  1. The discussion of the results is poorly written.

Authors’ response 1-1

Thank you for your helpful comment. In line with your comment, the theoretical background is further clarified. The small sample is added to the limitation. The methods section and discussion section are now strengthened.

Reviewer 1’s comment 2

  1. “Accordingly, this cross-sectional study aimed to evaluate the relationships between mental wellbeing, academic motivation (intrinsic motivation, extrinsic motivation and amotivation), self-compassion (self-reassurance) and self-criticism (self-inadequacy and self-hate)”.

  1. “Our findings can help educators to identify effective means to protect students’ mental wellbeing. Cultivating students’ self-compassion may be helpful to protect their mental wellbeing”. I can agree with that.

  1. Researchers conducted research in the Czech Republic. They represent different centers, therefore they can test a larger number of people. There are more places in the Czech Republic where you can do research.

  1. The authors selected too small a sample of respondents for the research, therefore the results and conclusions cannot be generalized (130 – 119). Statistical analysis on such a small sample is not correct.

Authors’ response 1-2

We have conducted a priori power analysis to calculate the required size. This is now clarified further.

Reviewer 1’s comment 3

5-6. Conclusions cannot be generalized. Too small sample was tested.

Authors’ response 1-3

This is addressed as above.

Reviewer 2 Report

Mental wellbeing of Czech university students: academic moti- 2 vation, self-compassion and self-criticism

This research is a scientific study that contributes to the field of health.

The presence of sub-headings in the introduction is remarkable for the readers to examine the research.

Materials and Methods section: Participants subtitle: The date range in which the data were collected from the students should be written.

Author Response

Response Letter

Manuscript ID: healthcare-1948142

"Mental wellbeing of Czech university students: academic motivation, self-compassion and self-criticism”

Dear Reviewers,

Thank you for your helpful feedback. We have systematically revised our manuscript addressing the points you have raised. Please see our responses below. We hope this revised paper is now acceptable for publication. We extend our sincere gratitude to you for your feedback that has significantly helped to strengthen the paper.

Reviewer 2

Reviewer 2’s comment 1

This research is a scientific study that contributes to the field of health.

The presence of sub-headings in the introduction is remarkable for the readers to examine the research.

Materials and Methods section: Participants subtitle: The date range in which the data were collected from the students should be written.

Authors’ response 2

Thank you for your helpful feedback. In line with your comment, now the recruitment information is added.

Reviewer 3 Report

This paper is well written. It evaluates the relationships between mental wellbeing, academic motivation (intrinsic motivation, extrinsic motivation and amotivation), 18 self-compassion (self-reassurance) and self-criticism (self-inadequacy and self-hate), which is interesting has important academic values. This paper is high quality in terms of research paradigm and English language. However, I have three minor points:

1. In your research methodology, you said: of which 119 completed 158 the measures. The students were Czech (n = 98) and Slovakian (n = 21), had an age range 159 between 19 to 44 years (M = 21.87, SD = 3.32), and consisted of 93 females and 20 males. My question is about the gender bias. Do you think it is an unbalance of the gender ratio? Two many female participants. I am worried about the unbalance. 

2.  In your research methodology, you mentioned: s. Overall, the number of participants reached the required sample size of 115 according to statistical power calculations. Could you explain the statistical power calculations? And the process of filter of 115 participants?

3. In conclusion, you only discussed the results and policy implications for Czech education. Could you provide more discussion on generalizability? Since you adopted a quantitative research design, generalizability is an important issue that you must consider. 

Author Response

Response Letter

Manuscript ID: healthcare-1948142

"Mental wellbeing of Czech university students: academic motivation, self-compassion and self-criticism”

Dear Reviewers,

Thank you for your helpful feedback. We have systematically revised our manuscript addressing the points you have raised. Please see our responses below. We hope this revised paper is now acceptable for publication. We extend our sincere gratitude to you for your feedback that has significantly helped to strengthen the paper.

Reviewer 3

Reviewer 3’s comment 1

This paper is well written. It evaluates the relationships between mental wellbeing, academic motivation (intrinsic motivation, extrinsic motivation and amotivation), 18 self-compassion (self-reassurance) and self-criticism (self-inadequacy and self-hate), which is interesting has important academic values. This paper is high quality in terms of research paradigm and English language. However, I have three minor points:

  1. In your research methodology, you said: of which 119 completed 158 the measures. The students were Czech (n = 98) and Slovakian (n = 21), had an age range 159 between 19 to 44 years (M = 21.87, SD = 3.32), and consisted of 93 females and 20 males. My question is about the gender bias. Do you think it is an unbalance of the gender ratio? Two many female participants. I am worried about the unbalance.

Authors’ response 3-1

Thank you for your helpful feedback. Indeed, that is what we wondered as well. The main question to be asked was whether findings from our sample are applicable to a more general population. Because our sample was psychology students, we referred to the national data about the general psychology students in the Czech Republic, and ensured that the representativeness was high. This is noted in the methods section.

Reviewer 3’s comment 2

  1. In your research methodology, you mentioned: s. Overall, the number of participants reached the required sample size of 115 according to statistical power calculations. Could you explain the statistical power calculations? And the process of filter of 115 participants?

Authors’ response 3-2

No problem. Now the details of our power calculation are added.

Reviewer 3’s comment 3

  1. In conclusion, you only discussed the results and policy implications for Czech education. Could you provide more discussion on generalizability? Since you adopted a quantitative research design, generalizability is an important issue that you must consider.

Authors’ response 3-1

In line with your comment, now generalisability is added to the discussion.

Reviewer 4 Report

Dear Authors,

Thank you for opportunity of reading your paper. This is very important topic,

in the context of the mental health of the population, and its  long-term consequences. In my opinion you did great job, and I recommend to publish. Congratulation!

Author Response

Response Letter

Manuscript ID: healthcare-1948142

"Mental wellbeing of Czech university students: academic motivation, self-compassion and self-criticism”

Dear Reviewers,

Thank you for your helpful feedback. We have systematically revised our manuscript addressing the points you have raised. Please see our responses below. We hope this revised paper is now acceptable for publication. We extend our sincere gratitude to you for your feedback that has significantly helped to strengthen the paper.

Reviewer 4

Reviewer 4’s comment 1

Dear Authors,

Thank you for opportunity of reading your paper. This is very important topic,

in the context of the mental health of the population, and its  long-term consequences. In my opinion you did great job, and I recommend to publish. Congratulation!

Authors’ response 4-1

Thank you for your kind words and helpful feedback.

Round 2

Reviewer 1 Report

-